# Memory Properties of Zr-Doped ZrO$_2$ MOS-like Capacitor

**Catalin Palade \*** [ID]**, Adrian Slav, Ionel Stavarache** [ID]**, Valentin Adrian Maraloiu** [ID]**, Catalin Negrila and Magdalena Lidia Ciurea** [ID]

National Institute of Materials Physics, 405A Atomistilor Street, 077125 Magurele, Romania
\* Correspondence: catalin.palade@infim.ro

**Abstract:** The high-k-based MOS-like capacitors are a promising approach for the domain of non-volatile memory devices, which currently is limited by SiO$_2$ technology and cannot face the rapid downsizing of the electronic device trend. In this paper, we prepare MOS-like trilayer memory structures based on high-k ZrO$_2$ by magnetron sputtering, with a 5% and a 10% concentrations of Zr in the Zr–ZrO$_2$ floating gate layer. For crystallization of the memory structure, rapid thermal annealing at different temperatures between 500 °C and 700 °C was performed. Additionally, Al electrodes were deposited in a top-down configuration. High-resolution transmission electron microscopy reveals that ZrO$_2$ has a polycrystalline–columnar crystallization and a tetragonal crystalline structure, which was confirmed by X-ray diffraction measurements. It is shown that the tetragonal phase is stabilized during the crystallization by the fast diffusion of oxygen atoms. The capacitance–voltage characteristics show that the widest memory window ($\Delta V = 2.23$ V) was obtained for samples with 10% Zr annealed at 700 °C for 4 min. The charge retention characteristics show a capacitance decrease of 36% after 10 years.

**Keywords:** non-volatile memory; high-k ZrO$_2$; MOS-like capacitor; charge retention





## 1. Introduction

In recent decades, the miniaturization trend of electronic devices has been the mainstream approach for increasing computational power and data storage capabilities. The usage of SiO$_2$ as a dielectric oxide for classical metal–oxide–semiconductor (MOS) devices [1,2] was made possible due to SiO$_2$'s very good electrical stability and native interface with Si. The scaling down of the SiO$_2$ tunneling oxide was the preferred method used for increasing the density of the memory structures on the surface unit. However, this approach has some limitations as the SiO$_2$ tunneling oxide has to be thinned down to 2 nm [3]. If the value is below this, the process of direct tunneling of the charge carriers between the Si substrate and the floating gate layer through the SiO$_2$ tunneling layer leads to a high-leakage current and a low-charge retention capability of the MOS-like capacitor [4]. A solution to this problem is to replace the SiO$_2$ with a high-k oxide, such as ZrO$_2$, HfO$_2$ or Al$_2$O$_3$ [5–11], that has a dielectric constant of $k > 20$, which is sufficiently high to have a low equivalent oxide thickness (EOT) [12] and low enough to avoid the lateral fringing-field effect [13,14] to be used in MOS-like capacitors. ZrO$_2$ is characterized by a good interface with Si [15] and being thermally stable [16], and the temperature of the phase transformation from monoclinic to tetragonal is almost 500 °C, which is lower than that for HfO$_2$ [17,18]. Additionally, the dielectric constant, k, of the tetragonal phase is higher than 40 [19]. It is important to have a lower phase transition temperature, as using lower temperature processing together with the magnetron sputtering (MS) deposition method (which is a cheap and reproducible method) means cost-effective production and equipment investments.

In the literature, there are very few articles related to the memory properties of the ZrO$_2$-based trilayer MOS-like capacitors. Some of these articles describe the charge storage properties of Ge or Zr nanocrystals embedded in the ZrO$_2$ matrix as a floating gate

layer [20–22] located at a tunnelable distance from the Si substrate. Other articles describe the ferroelectric properties of $ZrO_2$ in association with $HfO_2$ [23–25].

In this paper, we use Zr-doped $ZrO_2$ for the formation of charge storage centers in a $ZrO_2$-based MOS-like trilayer structure. The main advantage of using an oxygen-deficient $ZrO_2$ layer as a floating gate layer compared with Zr nanocrystals embedded in $ZrO_2$ is related to the high density of the storage centers. This is translated into a very stable memory structure with a long charge retention time. For this, we prepared a MOS-like memory structure of *ZrO₂ [gate oxide]/Zr–ZrO₂ [floating gate]/ZrO₂ [tunneling oxide]/p-Si* by MS, followed by rapid thermal annealing (RTA) between 500 °C and 700 °C and a metallization process. In the Zr–$ZrO_2$ floating gate layer, we used 5% and 10% Zr. The broadest memory window (ΔV = 2.23 V) was obtained from the samples with 10% Zr annealed at 700 °C.

## 2. Materials and Methods

### 2.1. Sample Preparation

The memory structures *ZrO₂ [gate oxide]/Zr–ZrO₂ [floating gate]/ZrO₂ [tunneling oxide]/p-Si* were prepared by using an MS deposition technique (Surrey Nanosystems Gamma 1000 equipment, East Sussex, UK) on a *p*-type Si substrate (1–10 Ω·cm resistivity). Before the deposition, the Si wafers were chemically cleaned by using a two-step procedure. In the first step, the organic contaminants were removed in a piranha solution ($H_2O_2$:$H_2SO_4$ = 1:3), followed by the second step of native $SiO_2$ removal in an aqueous solution of HF (2%). The preparation of the trilayer MOS-like capacitor starts with the deposition of a 13-nm tunneling $ZrO_2$ layer from a $ZrO_2$ target on which the power of P = 60 W RF was applied. Then, a 13-nm Zr-doped $ZrO_2$ floating gate layer was deposited by co-sputtering the Zr and the $ZrO_2$ (P = 60 W RF) from separate targets. For the Zr, we used two concentrations of 5% and 10% (P = 10–30 W DC) for obtaining 2 types of samples. Finally, the gate oxide layer of 30 nm was also deposited at a P = 60 W RF. The memory structure deposition was performed in an Ar atmosphere at a 25-sccm (standard cubic centimeters per minute) flow rate and a working pressure of p = 3 mTorr. To ensure the uniformity of the layer deposition, the Si substrate was rotated at 15 rpm. Additionally, a 56-nm $ZrO_2$ control structure was deposited under similar conditions as the trilayer one. After the deposition, the trilayer memory structure was nanostructured by RTA at different temperatures between 500 °C and 700 °C for 4 min in an $N_2$ atmosphere.

Table 1 presents the structure sketch and the preparation details of the trilayer MOS-like capacitor and the control structure.

**Table 1.** Structure configuration and preparation details of the trilayer *ZrO₂ [gate oxide]/Zr–ZrO₂ [floating gate]/ZrO₂ [tunneling oxide]/p-Si* and the control structure.

| | Name | Floating Gate Layer Composition | Structure | Layers Thickness [nm] | Thermal Annealing |
|---|---|---|---|---|---|
|  | Zr5-500 Zr5-600 Zr5-700 Zr10-500 Zr10-600 Zr10-700 | 5% Zr + 95% $ZrO_2$<br><br>10% Zr + 90% $ZrO_2$ | $ZrO_2$/Zr-$ZrO_2$/$ZrO_2$/Si | 30/13/13 | T = 500 °C, t = 4 min<br>T = 600 °C, t = 4 min<br>T = 700 °C, t = 4 min |
|  | Control | - | $ZrO_2$/Si | 56 | T = 700 °C, t = 4 min |

For the electrical characterization, Al electrodes with a 1-mm² area were deposited by thermal evaporation in a top-down configuration, forming a MOS-like capacitor.

*2.2. Characterization Methods*

The trilayer MOS-like capacitor structure and morphology were characterized by high-resolution transmission electron microscopy (HRTEM) in the Jeol JEM-ARM 200F electron microscope (JEOL, Tokyo, Japan). The crystalline structure was determined by X-ray diffraction measurements using the Rigaku SmartLab diffractometer (Rigaku, Tokyo, Japan), and the oxidation states of Zr were determined by X-ray photoelectron spectroscopy (XPS). A SPECS XPS spectrometer (SPECS Surface Nano Analysis GmbH, Berlin, Germany) with a PHOIBOS 150 analyzer was used for this. This uses a 300-W monochromatic RX source (Al Kα—1486.61 eV) and a SPECS FG15/40 (SPECS Surface Nano Analysis GmbH, Berlin, Germany) flood gun for charge neutralization.

The memory characteristics of each trilayer structure were determined by capacitance–voltage (*C*–*V*) measurements using an Agilent 4980A LCR meter (Agilent Technologies, Santa Clara, CA, USA). The frequency dependence of the capacitance (*C-f*) was also measured between 10 kHz and 1 MHz with different voltages applied to the gate electrode. The capacity of the MOS-like trilayer capacitor to retain electrical charges was determined by the time dependence of the capacitance (*C-t*), measured at a frequency of 1 MHz.

## 3. Results and Discussion

The structure and morphology of the sample with the 10% Zr content in the floating gate layer annealed at 700 °C for 4 min (Zr10-700) were determined by cross-section HRTEM analyses (Figure 1a), where the columnar crystallization of $ZrO_2$ can be observed.

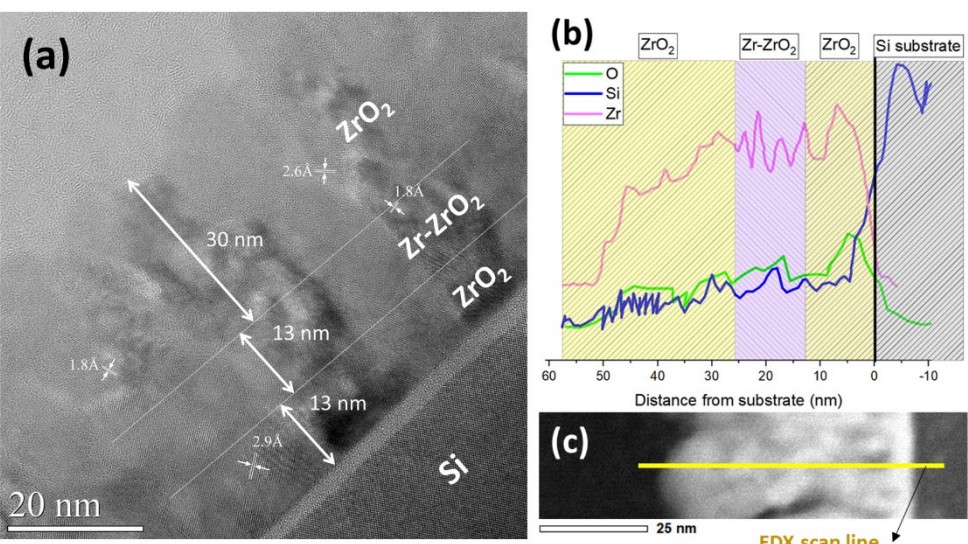

**Figure 1.** (**a**) HRTEM image obtained of the Zr10-700 structure; (**b**) EDX analysis with relative concentrations of Zr (magenta), Si (blue), and O (green), measured on the cross-section of the sample; (**c**) High-angle annular dark-field (HAADF) cross-section image.

In the HRTEM image, it is possible to measure some lattice fringe distances of 0.29 nm, 0.18 nm, and 0.16 nm that correspond to the planes (111), (002), and (103) of the $ZrO_2$ tetragonal phase, respectively.

The contrast between the deposited layers is low, and the trilayer configuration of the memory capacitor cannot be observed as distinct individual layers. The reason for this is the presence of the columnar crystallization process. The fast diffusion of oxygen atoms leads to uniform nonstoichiometric suboxides in the whole trilayer structure, not only in the middle layer. This will stabilize the tetragonal phase during the crystallization. In our opinion, the crystallization process begins with the nucleation on the tunneling layer, and the crystal growth continues normally to the free surface in a columnar morphology with a growth rate that depends on the nuclei orientations.

Some information about the structural configuration of the trilayer structure is obtained by analyzing the energy-dispersive X-ray spectroscopy (EDX) data. Figure 1b presents the relative concentration of Zr (magenta), Si (blue), and O (green) measured on the cross-section of the Zr10-700 sample (Figure 1c).

The maximum concentration of Zr is located near the floating gate layer, as expected from the preparation conditions. These results confirm the trilayer structure configuration of the MOS-like memory capacitor.

The polycrystallinity of the trilayer memory structure observed in the HRTEM analysis (Figure 1a) was also confirmed by the two-dimensional fast Fourier transform (FFT) (Figure 2b) applied to the HRTEM image (Figure 2a), followed by an inverse FFT (IFFT) (Figure 2c). The colored regions are obtained by filtering the spatial frequencies and orientations in the FFT image, which are labeled with numbers and colors in Figure 2b. Each colored region in Figure 2c represents a $ZrO_2$ nanocrystallite or a group of them having the same lattice fringe spacing and direction, giving the same spot in the FFT image. For example, the green area shows a group of four nanocrystallites that have the same crystalline orientation and similar lattice fringes in the HRTEM image. Each nanocrystallite resembles a coherent domain of $ZrO_2$.

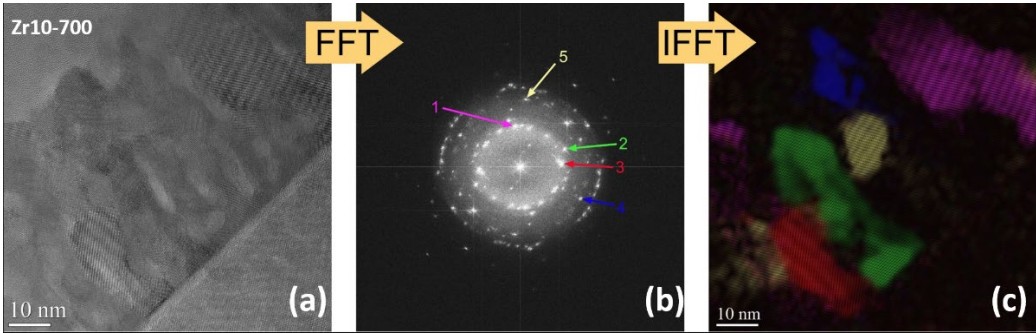

**Figure 2.** (**a**) HRTEM image obtained on the Zr10-700; (**b**) FFT algorithm applied on the HRTEM image. Each number correspond to a different lattice fringe spacing and direction; (**c**) IFFT image with colored regions that represents $ZrO_2$ nanocrystallites with different crystalline lattice fringe spacings and orientation.

The crystalline structure of the MOS-like capacitor was also evidenced by XRD measurements (Figure 3).

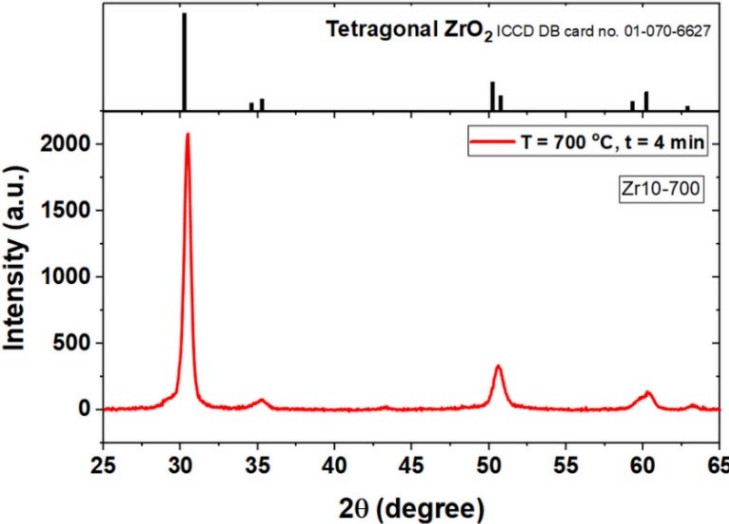

**Figure 3.** XRD spectra of the sample obtained after 4 minutes at 700 °C RTA (Zr10-700).

The XRD diffractogram presents high intensity peaks positioned at 2θ = 30.48 deg, 2θ = 35.2 deg, 2θ = 50.6 deg, and 2θ = 60.32 deg. These peaks correspond to a high-pressure tetragonal phase of the $ZrO_2$, indexed in the ICCD DB card no. 01-070-6627, for the crystallographic planes of (111), (002), (200), and (211), respectively. The 30.48 deg peak corresponds to the (111) plane with a lattice spacing of 0.293 nm, which is smaller than the 0.295 nm (for 30.28 deg) of the tetragonal lattice phase. However, the experimental (111) peak has a very large base (from 28.4 to 31.5 deg) covering the positions of the (-111) and (111) peaks of the monoclinic phase of the $ZrO_2$. This monoclinic phase is present in several percentages in the structure and explains the small coherence size of the $ZrO_2$ crystallites.

At room temperature and atmospheric pressure, the most stable phase of $ZrO_2$ is the monoclinic phase, which is characterized by a lower value of the dielectric constant compared with the tetragonal phase. In our case, by co-depositing between 10% Zr–90% $ZrO_2$ for the floating gate layer, some amorphous zones of the $ZrO_2$ at the nanometric scale will probably appear.

A possible explanation for the tetragonal phase being the majority phase in the trilayer structure is that during the RTA at 700 °C, the crystallization starts in both the tunneling $ZrO_2$ and the gate $ZrO_2$ and continues in the floating gate layer, similarly to the case of $HfO_2$-based trilayers. [26]. In the present case, the internal strain appears by oxygen vacancy formation and is also due to the 10% Zr excess in the floating gate.

We also performed X-ray photoelectron spectroscopy (XPS) measurements at the surface of the trilayer structure (Figure 4).

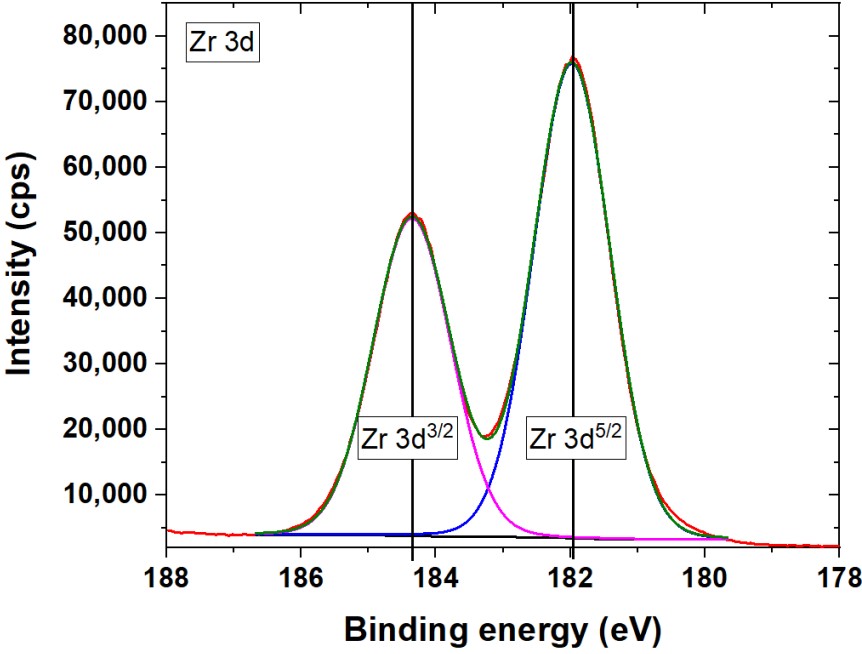

**Figure 4.** XPS spectra of the Zr 3d obtained on the Zr10-700. The colored lines represent components of spectral deconvolution (Voigt profiles) were each maximum corresponds to Zr 3d doublets.

As it can be seen, the Zr 3d doublets Zr $3d^{5/2}$ and Zr $3d^{3/2}$ have a splitting energy of 2.4 eV and an area ratio of 3/2. The binding energy clearly indicates that Zr is in an oxidation state; therefore, there is no metallic Zr in the $ZrO_2$ gate oxide layer contributing to the formation of the tetragonal phase of $ZrO_2$.

The memory properties of the MOS-like capacitor with Zr concentrations of 5 and 10% in the floating gate layer and annealing temperatures between 500 °C and 700 °C (for 4 min) were investigated by *C–V* measurements. The *C–V* characteristics are presented in Figure 5 together with the *C–V* curve taken on the control structure, for comparison.

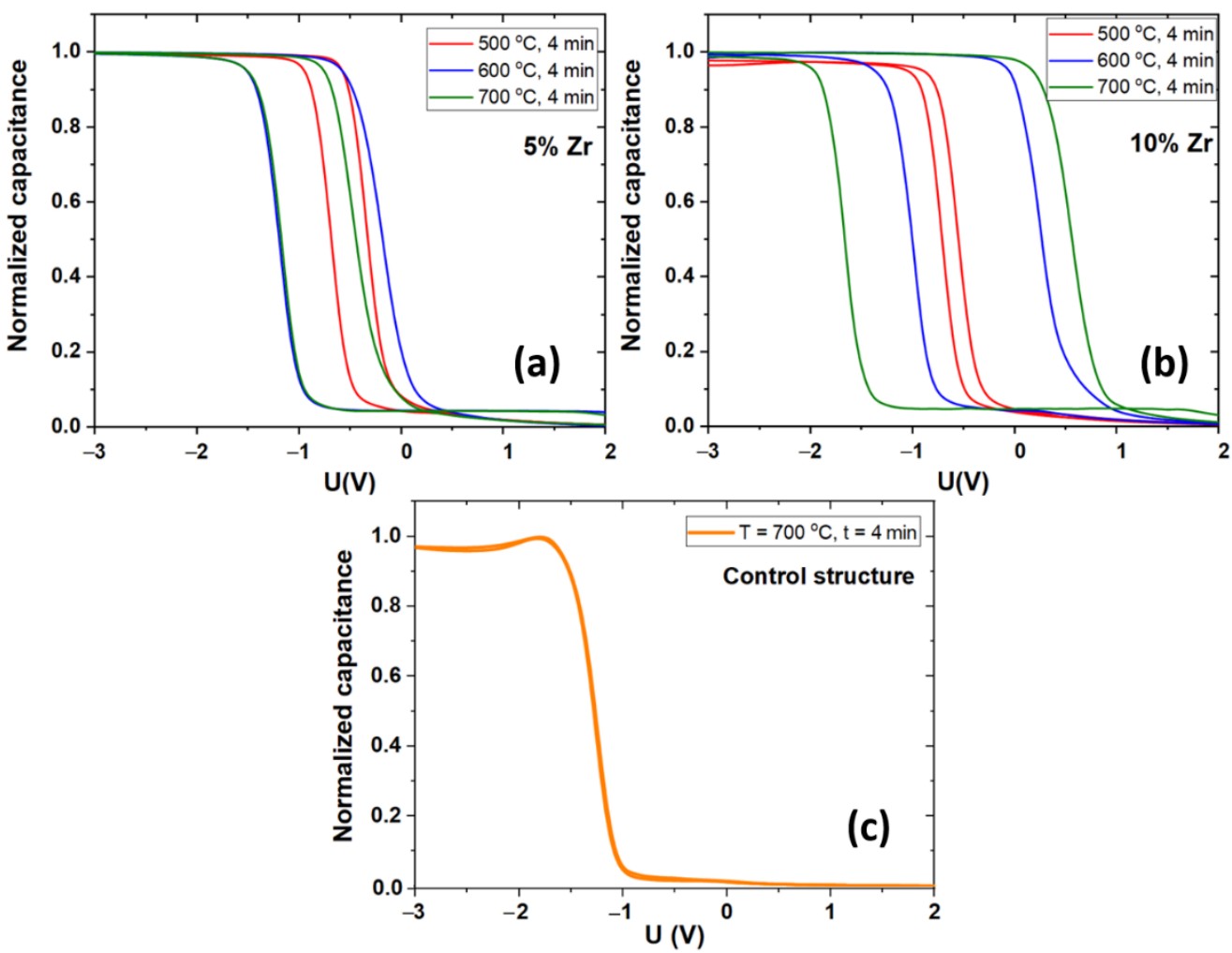

**Figure 5.** *C–V* characteristics obtained on the MOS-like capacitor with (**a**) 5% and (**b**) 10% concentration in the floating gate layer, annealed at 500°C, 600 °C, and 700 °C for 4 min; (**c**) *C–V* curves obtained on the control structure of 56 nm of $ZrO_2$ annealed at 700 °C for 4 min.

The *C-V* measurements were performed by swiping the voltage between +2 V (inversion regime) and −3 V (accumulation regime) with a swiping speed of 1 V/s at 1 MHz frequency [27]. The hysteresis loops in Figures 5a and 5b show that the best memory window of ΔV = 2.23 V is obtained on the capacitors with the 10% Zr content in the floating gate and an RTA at 700 °C (for 4 min in an $N_2$ atmosphere), in contrast with the *C–V* characteristic of the control structure (having no hysteresis).

Therefore, we can conclude that the memory properties of the trilayer MOS-like capacitor are due to both oxygen vacancies present in the trilayer structure and to Zr excess in the $Zr–ZrO_2$ floating gate layer acting as charge storage nodes.

The sample with the best memory characteristics (Zr10-700) was chosen for further investigations of the frequency dependence of the capacitance (Figure 6a) and the resistance (Figure 6b). The *C-f* and *R-f* characteristics were measured at different voltage values (−3 V–+2 V) applied to the $ZrO_2$ gate oxide electrodes. The capacitance and resistance values were recorded during the frequency swiping between 10 kHz and 1 MHz (Figure 6).

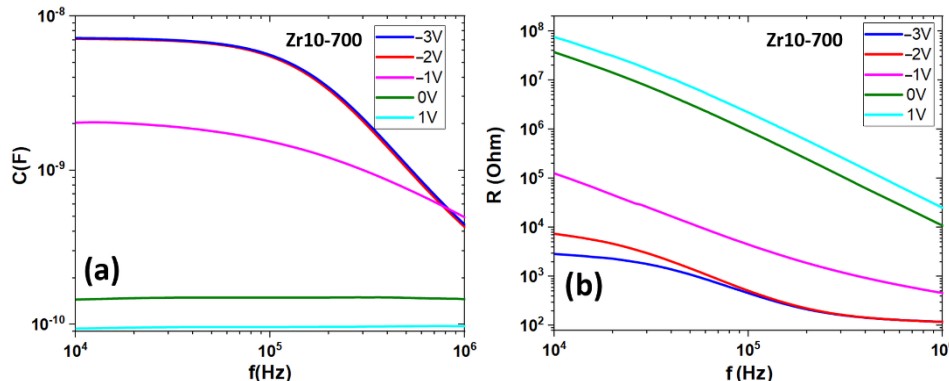

**Figure 6.** (**a**) *C-f* and (**b**) *R-f* measurements made at different voltages applied to the gate oxide electrode.

The voltage dependence of the *C-f* and *R-f* characteristics can be explained by the voltage dependence of the depletion region at the interface between the Si substrate and the $ZrO_2$ tunneling layer. By applying a negative voltage on the $ZrO_2$ gate oxide, the majority of charge carriers (holes) accumulate at the Si–$ZrO_2$ interface, meaning that there is no depletion region at the Si surface [27]. In the present case, the total capacitance is at its maximum value and is given only by the capacitance of the trilayer structure. By swiping the applied voltage from negative to positive, the Si–$ZrO_2$ interface is depleted of charge carriers. In this case, the total capacitance is the series capacitance of the trilayer structure and the capacitance of the depleted region. Since the depleted region capacitance depends on the dielectric constant of the Si and the width of the depleted region, the total capacitance decreases as the voltage approaches the inversion regime, where the minority of charge carriers arrive at the Si–$ZrO_2$ interface due to the high electric field.

In our previous work [28], we studied the frequency dependence of the capacitance and the resistance of a trilayer structure and developed a model that allows us to simultaneously fit the *C-f* and *R-f* characteristics. From the fit, we extracted the material parameters, such as the dielectric constant, of each layer in the MOS-like capacitor. We applied the same model to this structure ($ZrO_2$ *[gate oxide]*/Zr–$ZrO_2$ *[floating gate]*/$ZrO_2$ *[tunneling oxide]*/p-Si) and fitted the frequency dependence of the capacitance and the resistance in the accumulation regime (−3 V) to determine the dielectric constant of each layer. The results of the fit are presented in Figure 7.

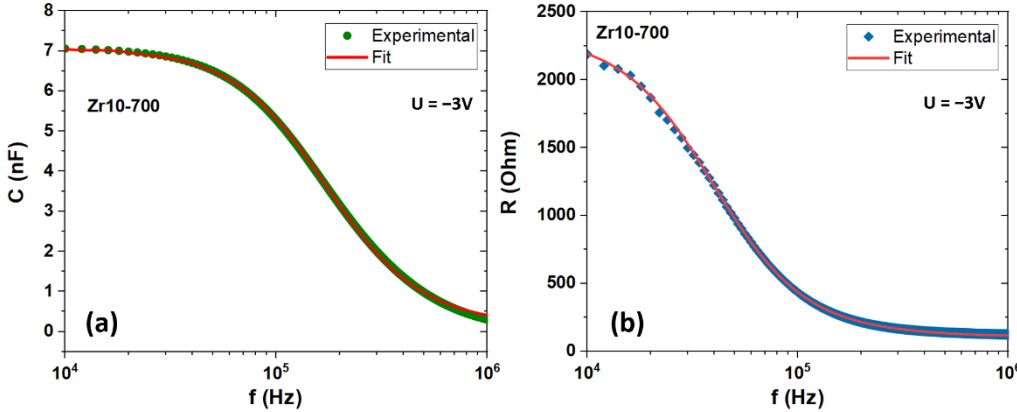

**Figure 7.** The frequency dependence of the (**a**) capacitance and (**b**) resistance measured on Zr10-700 in the accumulation regime (−3 V).

The values of the dielectric constants for each layer obtained from the *C-f* and *R-f* fits are presented in Figure 8.

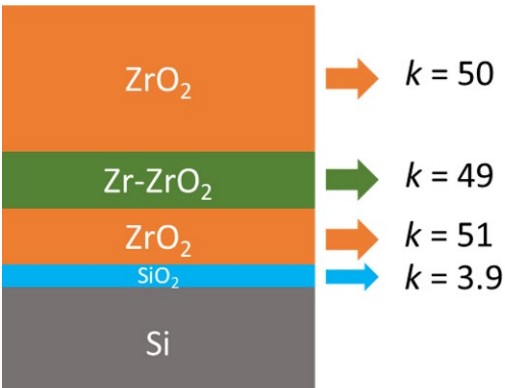

**Figure 8.** The schematic representation of the as-deposited trilayer structure and the dielectric constants obtained from the *C-f* and *R-f* fit curves from Figure 7 (the experimental *C-f* and *R-f* curves were measured on the Zr-10-700 structure).

Figure 8 shows the values of the dielectric constants obtained from the fit curves from Figure 7. These values of the dielectric constant correspond to the layers in the 700 °C RTA sample. One can observe that they are very close to each other and that they are typical for the tetragonal phase range [19]. These results are in good agreement with the XRD and HRTEM results. Additionally, the small variation in the dielectric constant values from one layer to another one shows that there is an internal strain in the trilayer structure due to oxygen vacancy formation and Zr excess.

The charge retention characteristics were also measured on the Zr10-700 sample at a frequency of 500 kHz and at a temperature of 23 °C. In Figure 9, the *C-t* characteristics are presented for both the programmed and the erased processes. The programming process was performed by charging the trilayer structure at −3 V for 1 s applied on the gate oxide (accumulation regime) and then the capacitance value was recorded in time at the applied voltage of −1.6 V (flatband voltage). The erasing process was performed by applying +2 V to the gate oxide for 1 s (inversion regime) and then the capacitance value was recorded in time at the applied voltage of +0.6 V (flatband voltage).

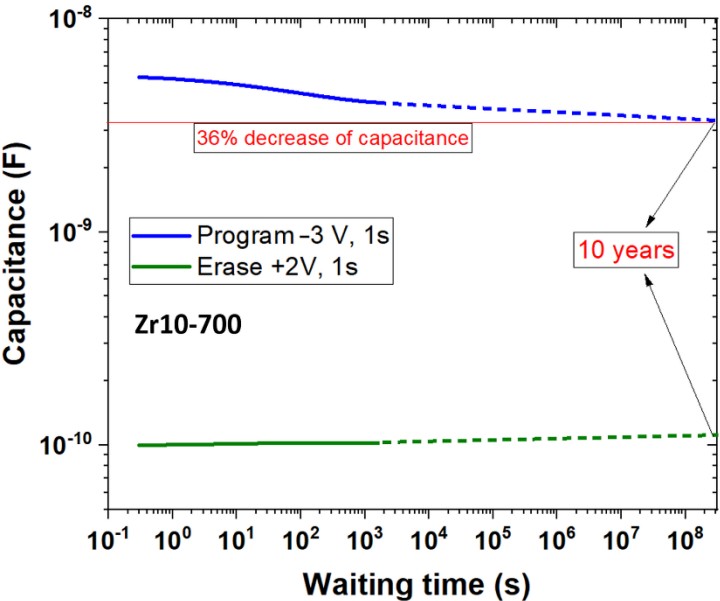

**Figure 9.** Charge retention characteristics measured at 500 kHz on the Zr10-700 sample. There is a 36% capacitance decrease after 10 years.

The capacitance-decreasing trend was extrapolated for 10 years, and we observed a 36% capacitance decrease, which is in the accepted range of a standard 50% capacitance decrease after 10 years for FLASH memory devices [29].

## 4. Conclusions

We prepared a MOS-like memory structure of *ZrO$_2$ [gate oxide]/Zr–ZrO$_2$ [floating gate]/ZrO$_2$ [tunneling oxide]/p-Si* using MS with a 5% and a 10% Zr content in the Zr–ZrO$_2$ floating gate layer, followed by an RTA process at 500 °C, 600 °C, and 700 °C. The trilayer structure Zr10-700 with the 10% Zr content in the floating gate layer is formed of polycrystalline ZrO$_2$ that is columnar crystallized in a tetragonal phase (HRTEM, XRD). The polycrystalline character of the trilayer memory structure is confirmed by FFT and IFFT analyses. We consider that the tetragonal phase is stabilized during the crystallization by the fast diffusion of oxygen atoms, leading to uniform nonstoichiometric suboxides in the whole trilayer. Additionally, XRD measurements show that besides the majority of the ZrO$_2$ tetragonal phase, the MOS-like capacitor structure contains several percentages of monoclinic phase that explains the small coherence size of the ZrO$_2$ crystallites. The XPS results show that Zr is in an oxidation state; therefore, there is no metallic Zr in the gate ZrO$_2$ layer to contribute to the ZrO$_2$ tetragonal phase.

The best memory properties were obtained on trilayer capacitors with 10% Zr annealed at 700 °C. The hysteresis loops taken on them have the broadest memory window of ΔV = 2.23 V, and the charge retention characteristics show a 36% capacitance decrease after 10 years. Additionally, the dielectric constants of each layer in the trilayer, obtained from the fit of *C-f* and *R-f* experimental curves, have values close to each other and they are typical for the tetragonal phase range. We consider that the most important contribution to the memory properties is given by oxygen vacancies with a high enough density, acting as charge storage centers.

**Author Contributions:** Conceptualization, methodology, software, C.P.; investigation, C.P., A.S., I.S., V.A.M., and C.N.; writing—original draft preparation, C.P.; writing—review and editing, C.P. and M.L.C.; supervision, C.P. All authors have read and agreed to the published version of the manuscript.

**Funding:** This research was funded by CNCS—UEFISCDI, projects no. PN-III-P1-1.1-PD-2019-1038, PN-III-P2-2.1-PED-2019-0205, PN-III-P2-2.1-PED-2019-4468, PN-III-P1-1.1-TE-2021-1537 within PNCDI III and by the Romanian Ministry of Research, Innovation, and Digitalization.

**Institutional Review Board Statement:** Not applicable.

**Informed Consent Statement:** Not applicable.

**Data Availability Statement:** Not applicable.

**Conflicts of Interest:** The authors declare no conflict of interest.

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
