# Peer review of "Memory Properties of Zr-Doped ZrO2 MOS-like Capacitor"

_coatings, doi:10.3390/coatings12091369_

Round 1

Reviewer 1 Report

The authors fabricated ZrO2/Zr-ZrO2/ZrO2/p-Si MOS-like capacitor with memory behaviors. Actually, it exhibits good memory properties, but I have several questions about the floating gate layer Zr-ZrO2. In addition, there are some mistakes in the manuscript. Details have been listed below.

1. HRTEM, FFT and IFFT images of Zr-ZrO2 in Figure 3 cannot reveal the Zr conditions in polycrystalline ZrO2. I don't know whether Zr atoms are doped in ZrO2 structures or Zr particles are forced to connect with ZrO2 nanocrystals.

2. FFT patterns are not the real space results. I prefer selected area electron diffraction (SAED) results to show the crystallinity, which can also be assigned to correlative oriented ZrO2 nanocrystals.

3. Band structures may be important in memory mechanism. The authors can provide some DFT results with band structure diagrams to explain why the device exhibits excellent memory performances.

4. One mistake in Page 4, Line 134-136. The description of peak positions is not consistent with the XRD results in Figure 4. The data are twice of the diffraction angles. A better way is to describe the distances of specific lattice facets rather than the diffraction angles.

Author Response

Thank you for reviewing our paper coatings-1916824 and for the valuable comments.

Comment 1. HRTEM, FFT and IFFT images of Zr-ZrO2 in Figure 3 cannot reveal the Zr conditions in polycrystalline ZrO2. I don't know whether Zr atoms are doped in ZrO2 structures or Zr particles are forced to connect with ZrO2 nanocrystals.

Response:  The images (including EDX results) presented in Figures 2 and 3 are taken on sample Zr10-700, i.e. ZrO2 [gate oxide] / 10% Zr- 90% ZrO2 [floating gate] / ZrO2 [tunneling oxide]/p-Si annealed by RTA at 700 0C, in which the floating gate was obtained by co-depositing Zr and ZrO2. Under these conditions we do not expect Zr nanocrystal formation, but we expect both the oxygen diffusion redistribution which leads to less stoichiometry of ZrO2 in the whole volume of the trilayer structure (not only in the floating gate) and the stabilization of the tetragonal structure (during crystallization).

Comment 2. FFT patterns are not the real space results. I prefer selected area electron diffraction (SAED) results to show the crystallinity, which can also be assigned to correlative oriented ZrO2 nanocrystals.

Response:  You are right about FFT. We prefer also SAED. However, our film thickness is much smaller than the minimum diameter of the SAED aperture. In fact, we have used in the paper the FFT pattern, only to show the different orientations of the coherent ZrO2 nanocrystals using the IFFT software.

Comment 3.  Band structures may be important in memory mechanism. The authors can provide some DFT results with band structure diagrams to explain why the device exhibits excellent memory performances.

Response:  We explain very good memory performances of trilayer MOS-like capacitor to be given by oxygen vacancies present in the trilayer structure and Zr-related states produced by Zr excess in Zr-ZrO2 floating gate layer, both of them acting as charge storage nodes. This is based on the fact that there is no metallic Zr in the trilayer (XPS and XRD), i.e. the trilayer is formed of nonstoichiometric ZrO2 (oxygen deficiency).

Comment 4. One mistake in Page 4, Line 134-136. The description of peak positions is not consistent with the XRD results in Figure 4. The data are twice of the diffraction angles. A better way is to describe the distances of specific lattice facets rather than the diffraction angles.

Response:  You are right, thank you. The text was corrected by replacing the theta with 2 theta.

Reviewer 2 Report

This work investigates experimentally the formation of charge storage centers in high-k oxide (ZrO2) trilayer structures for memory applications. The authors report on the fabrication of a MOS-like trilayer structure with various Zr doping levels and on its structural and electrical characterizations. A polycrystalline tetragonal phase is observed for the ZrO2 layer and memory window and charge retention characteristics are determined.

The realizations of high-k oxide MOS-like capacitors is relevant to the ongoing efforts in miniaturizing silicon-based components. The study carried out in this work shows promising results in this respect. The fabrication and characterization methods are well-described and the results would be worthy to be published in Coatings in my opinion.

Typos:
-"Section 2.1 Characterization methods" should probably be 2.2.
- "HAADF" is not defined in the caption of Fig. 2.

Author Response

Thank you for reviewing our paper coatings-1916824 and for the valuable comments.

The manuscript text was corrected accordingly. Thank you!

Reviewer 3 Report

1, The HRTEM image 2a shows the interface of ZrO2 and Zr-ZrO2 is unclear.

Even with the EDX analysis 2b, it's still hard to detect the interfaces. Can you

explain how you define the thickness of three layers? Thank you

2, The HRTEM image 2a also shows the non-uniformity of three layers. The lower ZrO2 layer has comparatively higher density compared with upper ZrO2 layer. Can you explain why?

3, Does the surface roughness affect the charge retention characteristics? The HRTEM image 2a shows the top ZrO2 layer is not smooth.

Author Response

Thank you for reviewing our paper coatings-1916824 and for the valuable comments. 

Comment 1. The HRTEM image 2a shows the interface of ZrO2 and Zr-ZrO2 is unclear. Even with the EDX analysis 2b, it's still hard to detect the interfaces. Can you explain how you define the thickness of three layers? Thank you.

Response:  The distances in the HRTEM image from Figure 2a are specified only to show the thicknesses of deposited layers (in Zr10-700 trilayer). No interface between layers is visible due to the columnar crystallization of the film. The trans-layers crystal growth process fades any traces of interface contrast between the deposited layers. The same happened in the EDX analysis. The 10%Zr concentration variation is in the range of the noise and also the thickness of the XTEM specimen varies from the film surface to the bottom. So practically, no interface between layers is visible in crystalized films.

Comment 2.  The HRTEM image 2a also shows the non-uniformity of three layers. The lower ZrO2 layer has comparatively higher density compared with upper ZrO2 layer. Can you explain why?

Response:  This is mainly due to the difference in thickness of the XTEM specimen at the film surface and bottom.

Comment 3. Does the surface roughness affect the charge retention characteristics? The HRTEM image 2a shows the top ZrO2 layer is not smooth

Response:  The film surface relief is also produced by columnar crystallization process. The surface roughness can affect the retention characteristics. In our case the trilayer surface is covered by the electrode and the interface between them it seems to be good enough, without local states deteriorating the retention property.   

Round 2

Reviewer 3 Report

Thanks for your detailed answers.